# Sodium Butyrate Enhances the Cytotoxic Effect of Etoposide in HDACi-Sensitive and HDACi-Resistant Transformed Cells

**DOI:** 10.3390/ijms242115913

**Published:** 2023-11-02

**Authors:** Olga O. Gnedina, Alisa V. Morshneva, Maria V. Igotti

**Affiliations:** Institute of Cytology, Russian Academy of Sciences, St. Petersburg 194064, Russia; olga.o.gnedina@gmail.com (O.O.G.); 1195alisa@gmail.com (A.V.M.)

**Keywords:** mitochondria-mediated apoptosis, histone deacetylase inhibitor, sensitization of transformed cells, Ku70, Bax protein–protein interactions

## Abstract

To overcome the problem of antitumor agent toxicity for normal cells, a combined therapy using drugs with synergistic effects seems to be more effective. We investigated the molecular mechanisms of the sensitization of tumor cells resistant and sensitive to histone deacetylase inhibitors (HDACis) upon etoposide treatment together with the HDACi sodium butyrate (NaBut). We showed that NaBut enhances the cytotoxic effect of etoposide in both HDACi-sensitive and HDACi-resistant cells due to the accumulation of the Bax protein and the dissociation of Ku70–Bax inhibitory complexes. In HDACi-resistant cells, NaBut causes the cytoplasmic accumulation of Bax dissociated from mitochondria in complexes with Ku70 proteins. The increased phosphorylation of the pro-apoptotic Bad protein due to the NaBut-induced activation of Erk and Akt kinases is one of the possible reasons for the accumulation of Bax in the cytoplasm. Despite the inactivation of Bax in HDACi-resistant cells, its accumulation in the cytoplasm upon NaBut treatment makes it possible to enhance the apoptotic response against agents activating the intrinsic pathway of apoptosis. Thus, HDACis involved in combined therapy mediate the sensitization of tumor cells to genotoxic drugs, regardless of the cells’ resistance to HDACis.

## 1. Introduction

Despite the development of new treatment strategies, there remains a high need to search for new approaches to antitumor therapy given the development of chemotherapy resistance, predisposition to tumor recurrence, and the high toxicity of the drugs used toward normal cells. Therefore, it is necessary to reduce the concentration of therapeutic agents to a safe level for healthy cells without reducing their cytotoxic efficacy against malignantly transformed cells.

DNA-damaging agents remain the most commonly used anticancer chemotherapeutic drugs, aimed at causing significant damage to cancer cells with a high proliferation rate compared with slowly dividing healthy cells [1]. Among the most prescribed DNA-damaging anticancer drugs worldwide that have been in clinical use for half a century is etoposide, an inhibitor of topoisomerase II. Etoposide prevents topoisomerase II from re-ligating broken DNA strands, which causes the accumulation of DNA double-strand breaks. Etoposide is used in the treatment of various types of cancer, such as testicular, prostate, bladder, stomach, and lung cancer. Since etoposide can also interfere with the growth of normal body cells, side effects may occur. To avoid excess side effects and potential toxicity, the combination therapy, where etoposide can be used at reduced concentrations, may be a useful approach.

Histone deacetylase inhibitors (HDACis) are considered promising agents of combined therapy that can increase the sensitivity of tumor cells to cytotoxic drugs [2,3].

Histone deacetylases (HDACs) catalyze the removal of acetyl groups from modified lysine residues located in the N-terminal tail region of the core histones H2A, H2B, H3, and H4. In addition, histone deacetylases can use not only histone proteins as substrates but also other cell proteins, including p53, NFkB, Ku70, etc. [4,5]. HDAC inhibitors induce the hyperacetylation of histone and non-histone proteins and chromatin relaxation. The facilitated access of transcriptional factors to DNA leads to the reactivation of suppressed gene expression.

HDAC inhibitors cause a variety of antiproliferative effects in transformed and tumor cells, like cell cycle arrest, autophagy, senescence [6,7], cell death (apoptosis), and differentiation depending on the cellular context, as well as in the mutations that caused the particular malignant transformation [8]. Given the powerful antiproliferative potential of HDAC inhibitors against cancer cells and the lack of toxicity in normal cells [9,10,11], these substances are currently the subject of intensive research as a promising anticancer therapy. Some HDAC inhibitors have already reached clinical use or are in phases II and III of clinical trials [2,12,13,14]. However, as noted above, not all types of tumor cells die under the HDACi treatment, which reduces the potential of these compounds as tumor monotherapy agents and raises the question of finding mechanisms to overcome the resistance of tumor cells to HDAC inhibitors.

A number of HDAC inhibitors have shown low toxicity by themselves; however, in combination with cytotoxic drugs, HDACis have resulted in some antitumor reactions in patients, for example, with the microtubule reorganization inhibitor docetaxel [15]. In contrast, vorinostat and romidepsin have shown drug toxicity in a significant number of patients [8,9]. To overcome the clinical disadvantages of HDAC inhibitors, it is necessary to elucidate the molecular mechanisms of action of HDAC inhibitors in addition to or due to HDAC inhibition, particularly in combination with other anticancer drugs.

In this work, we used sodium butyrate (NaBut), a naturally occurring short-chain fatty acid that is a by-product of carbohydrate metabolism in the gut. NaBut is one of the most widely studied HDACis [16]; it inhibits histone deacetylases of classes I and IIA, and its effect is often mediated through the Sp1/Sp3 transcription factors [17]. In this work, we investigated the molecular mechanisms underlying the enhancement of the cytotoxic effect of etoposide under combined treatment with the HDAC inhibitor NaBut, which makes it possible to decrease drug concentrations while also maintaining their effectiveness in HDACi-sensitive and HDACi-resistant tumor cells.

## 2. Results

### 2.1. Sodium Butyrate Enhances the Antiproliferative Potential of Etoposide

The antiproliferative effects of the topoisomerase II inhibitor etoposide alone or combined with the histone deacetylase inhibitor sodium butyrate were investigated using the MTT test in oncogene-transformed mouse embryonic fibroblasts (mERas cell line). mERas cells were exposed to increasing concentrations of etoposide (0.5–2 μM) for 24 h. The data obtained demonstrate that etoposide treatment resulted in a dose-dependent decrease in cell viability (Figure 1a). A combined treatment with etoposide and the HDAC inhibitor sodium butyrate (NaBut) reduced the viability of transformed mERas cells by about 50% more effectively than the effect of etoposide alone (Figure 1a). MTT data demonstrate that the cytotoxic effect of the combined action of etoposide and NaBut exceeds the sum of the cytotoxic effects observed under the separate action of the drugs. 

The coefficient of drug interaction (CDI) was calculated to determine whether the actions of etoposide with the HDAC inhibitor NaBut are synergistic. CDI was computed as CDI = AB/(A × B), where AB represents the proportion of cells that survived after treatment with a combination of two drugs, scaled to the control group (untreated cells), and A or B represents the proportion of cells that survived after the single drug treatment, scaled to the control group. When the value of CDI is <1, it indicates that the activities of the two drugs are synergistic (CDI values ranging from 0.7 to 0.9 correspond to a moderate synergism of drugs). The CDI value for the simultaneous effect of 3 μM etoposide and 4 mM NaBut is 0.7, signifying a synergistic effect of this combination.

Flow cytometry analysis was used to explore the effect of etoposide alone or etoposide in combination with NaBut on the cell cycle distribution. The treatment of mERas cells with a non-toxic concentration of NaBut (4 mM) for 24 h does not induce accumulation of cells with sub-diploid DNA contents (Figure 1b). Etoposide induces an increase in cell proportion with sub-diploid DNA, implying the induction of an apoptotic cell death program. The simultaneous action of etoposide with the HDAC inhibitor enhances the signs of apoptotic cell death (Figure 1b).

A DNA fragmentation assay was used to evaluate the induction of apoptosis. The electrophoresis of genomic DNA demonstrates that NaBut itself does not cause nucleosomal DNA fragmentation in transformed cells (Figure 1c). However, it intensifies the DNA fragmentation induced by etoposide. Thus, etoposide induces nucleosomal DNA fragmentation, which is enhanced by the combined action of etoposide with NaBut (Figure 1c).

Using the colony formation assay, the long-term anti-proliferation effect of etoposide, either alone or in combination with NaBut, was performed in mERas cells. As shown in Figure 1d, etoposide alone reduces the number of colonies that grew after 14 days by approximately 50%. On the other hand, NaBut, when administered alone, reduces the clonability of transformed cells by around 20%. Notably, the combination of etoposide with NaBut significantly inhibits colony formation compared to the effect of the drugs when used individually (Figure 1d).

All of the aforementioned findings strongly indicate that NaBut augments the cytotoxic effect of etoposide, leading to a substantial reduction in cell viability.

### 2.2. NaBut Sensitizes Human Cancer Cell Lines to Low Concentrations of Etoposide

To further clarify whether the enhancement of the cytotoxic effect of etoposide by NaBut is specific to the engineered mERas cell line or if HDAC inhibitors can sensitize other Ras-mutant tumor cells as well, we examined two tumor cell lines harboring mutations in the *ras* gene and having varying sensitivity to HDAC inhibitors: HCT116 and A549.

The MTT analysis data reveal that A549 adenocarcinoma lung cancer cells exhibit high resistance to the topoisomerase II inhibitor etoposide, requiring a concentration of over 200 μM to achieve a 50% reduction in cell viability (Figure 2a). However, the same 50% decrease in A549 cell viability can be accomplished by the combined action of a two orders of magnitude lower concentration of etoposide (5 μM) along with the HDAC inhibitor NaBut (Figure 2b). It is worth noting that NaBut, on its own, reduces the proliferation of A549 cells, inducing cellular senescence rather than apoptosis [18,19]. The MTT assay reveals that the combined action of etoposide and NaBut (Etop/NaBut) reduces the viability of A549 cells compared with the separate action of each drug. Flow cytometry data demonstrate that low concentrations of etoposide (5 μM) or NaBut alone do not result in the accumulation of A549 cells with sub-diploid DNA content (Figure 2c). However, the combined treatment of Etop/NaBut leads to the accumulation of A549 cells with sub-diploid DNA, indicating the induction of apoptosis. This suggests that NaBut sensitizes A549 cells to low concentrations of etoposide.

Similar results were obtained for HCT116 cells. MTT data reveal that low concentrations of etoposide fail to reduce the viability of HCT116 cells (Figure 2d). However, NaBut increases the sensitivity of HCT116 cells to etoposide. Data in Figure 2e,f demonstrate that while NaBut alone reduces the viability of HCT116 cells by inducing apoptosis, it also enhances the cytotoxic effect of low concentrations of etoposide when used in combination.

Summarizing the above, we can conclude that NaBut increases the sensitivity of Ras-mutant human cancer cells, HCT116 and A549, to etoposide.

### 2.3. Sodium Butyrate Enhances the Mitochondria-Mediated Apoptotic Pathway

To investigate the mechanisms underlying the inhibition of cancer cell growth induced by etoposide alone or in combination with NaBut, the effect of these treatments on the caspase pathways was examined.

The immunoblot data reveal that etoposide induces an accumulation of the active (cleaved) form of caspase-9, which is the initiator caspase of the intrinsic apoptotic pathway, as well as caspase-3, an executioner caspase. However, etoposide does not activate caspase-8 (Figure 3a), which is typically associated with the extrinsic apoptotic pathway. NaBut alone does not activate caspases 3, 8, and 9. However, the combined treatment with etoposide and NaBut leads to a more potent activation of caspase-9 and caspase-3. Therefore, the data suggest that etoposide activates the caspases responsible for the internal mitochondria-mediated pathway, and the combined treatment with etoposide and NaBut enhances the activation of caspases.

To investigate the role of mitochondria in apoptosis induced by etoposide alone or in combination with NaBut, we examined changes in mitochondrial membrane potential (∆Ψ). Healthy mitochondrial membranes maintain a difference in electrical potential between the interior and exterior of the organelle, referred to as the membrane potential (∆Ψ), which decreases during apoptosis as a result of mitochondrial membrane permeabilization. The potential-dependent dye TMRM accumulates in the active mitochondria with intact membrane in proportion to ΔΨ. Loss of mitochondrial membrane potential leads to a decrease in or disappearance of the TMRM fluorescent signal.

The histogram of TMRM fluorescence demonstrates that the treatment with etoposide leads to the accumulation of cells with reduced ∆Ψ (Figure 3b). Notably, NaBut alone does not cause a decrease in ∆Ψ in mERas cells. However, when combined with NaBut, there is a significant increase in the proportion of cells exhibiting reduced ∆Ψ, suggesting that the HDAC inhibitor NaBut enhances the mitochondria-mediated apoptotic pathway.

Primarily, proteins of the Bcl-2 family play a crucial role in regulating mitochondria-mediated apoptosis and the permeability of the outer mitochondrial membrane. Pro-apoptotic factors of the Bcl-2 family, such as Bax/Bak, form a complex of proteins that creates permeability transition pores encompassing both mitochondrial membranes. The anti-apoptotic protein Bcl-2 functions to sequester the Bax and Bak proteins, reducing pro-apoptotic signaling through the mitochondria [20]. As expected, etoposide alone or in combination with NaBut failed to induce the mitochondrial membrane permeability and a loss of ∆Ψ in cells overexpressing Bcl-2 (mERas-Bcl-2 cells) (Figure 3d). Immunoblotting data confirm that caspase-9 and caspase-3 remain unaltered under the drug treatment in cells with Bcl-2 overexpression (Figure 3c).

MTT assay data demonstrate that overexpression of anti-apoptotic Bcl-2 or its functional equivalent, adenoviral protein E1B 19K (E1A + E1B cells), reduces the antiproliferative effects of both etoposide (Figure 4a) and etoposide in combination with NaBut (Figure 4b). E1B 19K is a homolog of the anti-apoptotic protein Bcl-2. Despite their sequence similarity, E1B 19K and Bcl-2 employ different mechanisms to inactivate the pro-apoptotic proteins Bax and Bak. The differing impact of Bcl-2 or its homologue, E1B 19K, on cell survival during etoposide treatment likely arises from these distinct mechanisms of Bax protein inactivation.

The flow cytometry data demonstrate that the inhibition of mitochondrial membrane permeabilization, either by Bcl-2 overexpression or by treatment with the mitochondrial inhibitor rotenone, prevents the accumulation of cells with sub-diploid DNA induced by etoposide alone or in combination with NaBut (Figure 4c).

We also utilized inhibitors of mitochondrial membrane permeabilization, such as cyclosporin A and rotenone, to study the role of mitochondria in cell death induced by etoposide. To assess the impact of cyclosporin A and rotenone on the modulation of etoposide cytotoxicity, we used etoposide at a concentration of 200 μM for HCT116 and A549 cells, and at 2.5 μM for mERas cells, corresponding to the lowest concentrations of etoposide that reduce cell viability. The MTT assay data demonstrate that chemical inhibitors of mitochondrial membrane permeability transition, such as cyclosporin A or rotenone, reduce etoposide-induced apoptosis in both HDACi-sensitive HCT116 cells (Figure 4f) and HDACi-resistant mERas and A549 cells (Figure 4d,e). All of these data confirm the activation of the mitochondria-mediated apoptotic program following treatment with etoposide in both HDAC-sensitive and HDAC-resistant cells.

### 2.4. HDAC Inhibitor Sodium Butyrate Affects the Expression of the Mitochondrial Apoptosis Pathway Regulators

To investigate the underlying mechanism of the enhanced pro-apoptotic effect of etoposide in combination with NaBut, we assessed the expression of apoptosis-related proteins. Specifically, we examined the expressions of Bcl-2 and Bax in both HDACi-resistant (mERas and A549) and HDACi-sensitive (HCT116) cells following treatment with 4 mM NaBut for 24 h. The data in Figure 5a demonstrate that NaBut is unable to modulate the expression of the Bcl-2 anti-apoptotic protein, but increases the level of pro-apoptotic Bax protein in all cell lines. The accumulation of Bax in both HDACi-resistant cells and cells where HDACis induce apoptosis indicates that HDACi-induced accumulation of Bax alone is not the decisive factor in the regulation of HDACi-dependent apoptosis. The immunoblotting data from the fractionated extracts of HDACi-resistant cells reveal that the accumulated Bax protein is localized in the cytoplasm and is dissociated from the mitochondria, rendering it unable to carry out pro-apoptotic functions (Figure 5b). These findings are consistent with the results showing that NaBut is incapable of inducing mitochondrial potential reduction and apoptosis in mERas cells (Figure 2 and Figure 4).

To reveal the molecular mechanisms of NaBut-mediated Bax activation in HDACi-sensitive cells and Bax inactivation in the cytoplasm of HDACi-resistant cells, we studied Bax protein interactions in NaBut-treated cells. Ku70 is a nuclear repair protein that also functions in cytoplasm and is known to be involved in retaining Bax interactions [21]. Figure 5c illustrates that the interaction of Bax with Ku70 is not reduced but, rather, enhanced in mERas and A549 cells upon NaBut treatment. In contrast, in HCT116 cells treated with NaBut, the Bax–Ku70 complexes dissociate. These results suggest that Bax interacts with Ku70 in both HDACi-sensitive and HDACi-resistant cells, and this interaction is disrupted by NaBut treatment only in HDACi-sensitive HCT116 cells.

To test whether Ku70 acetylation is responsible for the change in the Bax–Ku70 interaction, we immunoprecipitated extracts from mERas and HCT116 cells using an anti-acetyl-lysine antibody and subsequently performed Western blotting with antibodies to Ku70. The data presented in Figure 5c reveal that, in both cell types (HDACi-sensitive HCT116 and HDACi-resistant mERas and A549), Ku70 becomes acetylated following NaBut treatment. Notably, the NaBut-induced acetylation of the Ku70 protein in HDACi-resistant cells is considerably weaker compared with the level of Ku70 acetylation in HDACi-sensitive cells (Figure 5c).

Our results indicate that in HDACi-resistant cells, factors other than Ku70 acetylation may be involved in the regulation of Bax–Ku70 interaction, since the weak Ku70 acetylation is not sufficient to dissociate the Bax–Ku70 complexes or activate Bax in these cells.

The anti-apoptotic family members Bcl-2 and Bcl-xL prevent Bax from incorporating into the mitochondrial membrane, a process that can be inhibited by the Bad protein. The activity of Bad, which is regulated by the Ras-mitogen-activated protein kinase pathway [22,23], was investigated under HDACi treatment. Western blot data in Figure 6 indicate that NaBut increases the phosphorylation of the Bad protein and activates the upstream effector kinases Erk and Akt in HDACi-resistant mERas and A549 cells (Figure 6a). This prevents Bax from moving from the cytoplasm to the mitochondria. Conversely, In HCT116 cells, which undergo apoptotic death under NaBut treatment, there is no increase in the phosphorylation of Erk and Akt kinases (Figure 6a).

Thus, our results demonstrate that in HDACi-resistant cells, NaBut also enhances Bad protein phosphorylation through the activation of Erk and Akt kinases. This, in addition to Ku70 acetylation, results in prevention of Bax protein relocalization into mitochondria and a reduction in its pro-apoptotic functions.

Thus, the HDAC inhibitor NaBut does not induce apoptosis in HDACi-resistant cells. Instead, it modulates the regulators of the mitochondrial apoptosis pathway, creating conditions to enhance the cytotoxic effect of agents that induce the intrinsic pathway of apoptosis.

## 3. Discussion

In the present study, we investigated the ability of HDACi sodium butyrate (NaBut) to enhance the cytotoxic effect of etoposide, with the aim of reducing drug concentrations while maintaining its effectiveness, in HDACi-resistant and HDACi-sensitive tumor cells. We have shown that NaBut enhances the antiproliferative potential of etoposide in both HDACi-sensitive and HDACi-resistant transformed cells.

HDACi-resistant mERas and A549 cells do not die upon treatment with NaBut alone; however, the effect of the combined action of NaBut and etoposide exceeds the toxic effect of etoposide alone (Figure 1 and Figure 3). HDACi-responsive HCT116 cells undergo apoptotic death under the action of NaBut; however, the cytotoxic effect of the combined treatment with etoposide and NaBut also exceeds the toxic effect of the drugs alone (Figure 3).

Our data suggest that etoposide activates a mitochondria-mediated apoptotic program that is abolished by inhibition of the mitochondrial membrane permeability transition with Bcl-2 or E1B overexpression, or by the treatment with cyclosporine A or rotenone. Similar to our result, previous reports have shown that etoposide causes mitochondrial damage followed by caspase-9 and -3 activation, leading to apoptosis in human lung epithelial cells A549. This effect was prevented by the overexpression of Bcl-2 [24]. Accordingly, our data suggest that NaBut enhances the cytotoxic effect of drugs that initiate the intrinsic mitochondrial apoptosis pathway in both HDACi-responsive and HDACi-sensitive cells. These experiments indicate the ineffectiveness of using etoposide in combination with adenovirus infection in antitumor therapy. This is because replication-deficient adenoviruses used for delivery in processes like vaccination and anticancer therapy typically lack only E1A, but encode the viral analog of the Bcl-2 protein, E1B [25].

We investigated the mechanism underlying the differences in HDACi-dependent regulation of cell death and the similarities in HDACi-mediated sensitization of cells to etoposide in both HDACi-resistant and HDACi-sensitive cells. The Bax protein is a key regulator of the mitochondria-mediated pathway of apoptosis. Bax, a pro-apoptotic protein of the Bcl-2 family, translocates into the mitochondria during apoptosis where it induces mitochondrial outer membrane permeability leading to the release of pro-apoptotic factors such as cytochrome c and SMAC/Diablo into the cytosol. To associate with the mitochondria, the Bax protein must undergo oligomerization. The prosurvival Bcl-2 proteins prevent Bax from oligomerizing and incorporating into the mitochondrial membrane, causing Bax to re-translocate from the mitochondria into the cytosol [20]. We have shown that NaBut-induced accumulation of the pro-apoptotic Bax protein leads to a decrease in mitochondrial potential and induces apoptosis in HDACi-responsive cells. HDACi-resistant cells avoid apoptosis due to the sequestration of accumulated Bax in the cytoplasm (Figure 5b). However, NaBut-induced Bax accumulation enhances etoposide-caused cell death in both HDACi-responsive and HDACi-resistant cells. This is consistent with other studies that report both apoptosis induced by HDACi monotherapy [26,27] and the enhanced sensitivity of tumor cells to genotoxic drugs when combined with HDACi, which are mediated by increased Bax expression and its translocation into the mitochondria [3,28]. Accordingly, Bax accumulation is the determining mechanism of HDACi-mediated cell sensitization to etoposide, regardless of cell sensitivity to HDACi. Therefore, the study of the mechanisms of Bax protein retention in the cytoplasm of HDACi-resistant cells may indicate new potential targets for complex antitumor therapy.

Ku70 is a DNA repair protein that has been shown to suppress apoptosis by isolating Bax from mitochondria [21,29]. Ku70 also stabilizes Bax by preventing its ubiquitination [29]. Our studies demonstrated that Ku70 can directly interact with Bax in cytosol, thus preventing the apoptotic translocation of Bax to the mitochondria, both in HDACi-sensitive and HDACi-resistant cells (Figure 5c). In HDACi-sensitive cells, treatment with NaBut results in Ku70 acetylation and dissociation of the Bax–Ku70 complex, allowing Bax to translocate to the mitochondria and carry out its pro-apoptotic functions. However, in HDACi-resistant cells, weak acetylation of the Ku70 protein is insufficient to dissociate the Bax–Ku70 complex. Furthermore, the treatment of HDACi-resistant cells with NaBut leads to an increased association of Bax with Ku70.

It is known that acetylation of the Ku70 by CBP and PCAF acetyltransferases disrupts the interaction of Ku70 with the Bax protein, which leads to the translocation of Bax into the mitochondria [21,30]. In patients with glioma, a reduced level of Ku70 protein acetylation significantly correlates with tumor progression and reduced survival. This is due to the stabilization of the Bax–Ku70 interaction [31]. Accordingly, it can be assumed that the HDACi-induced increase in Ku70 acetylation may lead to the release of Bax and its translocation into the mitochondria. Similar to our result, previous studies have reported that HDACi-induced acetylation of Ku70 was accompanied by subsequent dissociation of Bax from Ku70 and apoptotic death in various cancer cells, including neuroblastoma [4,30], human lung cancer cell lines [32], and HeLa [21,32]. In other words, the modulation of the Bax–Ku70 interaction through the acetylation status of Ku70 may influence cellular apoptosis.

We showed that NaBut significantly increased Ku70 protein acetylation in HDACi-sensitive HCT116 cells, and this effect was more pronounced than in HDACi-resistant mERas and A549 cells (Figure 5c). Additionally, a decrease in protein acetylation was observed in HDACi-resistant T-cell lymphoma cells following treatment with the HDACi belinostat, as compared with the levels of protein acetylation in sensitive cells [33]. In some cases, HDACi-induced Ku70 acetylation does not lead to the release of Bax from Ku70 [4]. Similar to our result, it has been reported that in cells resistant to HDACi-induced apoptosis, the Ku70 protein is acetylated under treatment with SAHA or TSA but does not dissociate from the complex with the Bax protein [4]. These findings suggest that Ku70 acetylation is not the sole mechanism regulating Bax release from the Bax–Ku70 complexes in the cytosol; an additional stimulus is required to release Bax from the complex with Ku70.

Anti-apoptotic protein Bcl-2 prevents Bax oligomerization and incorporation into the mitochondrial membrane. A pro-apoptotic member of the Bcl-2 family Bad displaces Bax from binding to Bcl-2 or Bcl-XL, allowing Bax to oligomerize and integrate into the mitochondrial membrane. Phosphorylation of Bad, either through the Erk-regulated kinase cascade or the Akt cascade, abolishes the pro-apoptotic activity of Bad. This reduces its ability to bind with Bcl-2 or Bcl-XL and degrade the Bcl-2 complex with Bax [23]. Our data show that in HDACi-resistant cells, NaBut activates Erk and Akt kinases, leading to the phosphorylation of the Bad protein, which can result in the retention of inactive Bax in the cytoplasm. In contrast, in HDACi-sensitive cells, phosphorylation of Erk and Akt kinases does not increase upon treatment with NaBut (Figure 6). Similar to our result, it has been reported that the genotoxic drug cisplatin enhances Akt kinase activity more effectively in cisplatin-resistant ovarian cancer cell lines (Caov-3) compared with cisplatin-sensitive A2780 cells. It has been observed that downregulation of Akt sensitized the cells to cisplatin [23]. Accordingly, we have shown that the combined treatment with NaBut and the DNA-damaging agent etoposide reduces the activity of Erk and/or Akt kinases, thereby sensitizing HDACi-resistant tumor cells.

## 4. Materials and Methods

### 4.1. Cell Lines and Treatments

We used mouse embryonic fibroblasts that were transformed through calcium-phosphate transfection with an early region of human adenovirus type 5 (E1Aad5) in complementation with cHa-ras, which carries activating mutations at positions 12 and 61, to create the mERas cell line [34]. Bcl-2-overexpressing cells were obtained from mERas cells through calcium-phosphate transfection with plasmids pSFFV containing the human *bcl-2* gene and pSV2neo with a gene for resistance to geneticin. The A549 human lung cancer cell line and the human colorectal carcinoma cell line HCT116 were obtained from the Center for Collective Use “Collection of Vertebrate Cell Cultures”, supported by a grant from the Ministry of Education and Science of the Russian Federation (agreement No. 075-15-2021-683). Cells were cultivated in high-glucose Dulbecco’s modified Eagle’s medium (DMEM) with 10% fetal bovine serum (FBS, Hyclone, Cytiva, Washington, DC, USA) and gentamicin. Cells were treated with 4 mM sodium butyrate (NaBut) for 24 h, 2.5–300 μM etoposide, and 50 nM rotenone or 2 μM cyclosporine A (all chemicals were obtained from Sigma-Aldrich, St. Louis, MO, USA).

### 4.2. Cell Cycle Phase Distribution

Cells were permeabilized with 0.01% saponin at room temperature for 20 min. Subsequently, they were treated with 100 μg/mL RNAse A at 37 °C for 15 min and stained with 40 μg/mL Propidium Iodide. The probes were analyzed using a CytoFLEX flow cytometer (Beckman Coulter, Brea, CA, USA), with the dye detected in a 610/20 bandpass.

### 4.3. Immunoprecipitation and Immunoblotting

For immunoblotting, cells were lysed in a buffer containing 1% NP-40, 0.5% sodium deoxycholate, 0.1% SDS, 20 mM glycerophosphate, 1 mM sodium orthovanadate, 5 mM EGTA, 10 mM sodium fluoride, 1 mM phenylmethylsulfonyl fluoride, and a protease inhibitor cocktail (Roche, Sigma-Aldrich, St. Louis, MO, USA). For immunoprecipitation, cells were lysed in a buffer containing 10 mM Tris–HCl, pH 7.4, 150 mM NaCl, 0.5% Nonidet P-40, 1% Triton X-100, 20 mM—glycerophosphate, 1 mM sodium orthovanadate, 5 mM EGTA, 10 mM sodium fluoride, 1 mM phenylmethylsulfonyl fluoride, and a protease inhibitor cocktail. Prior to immunoprecipitation, cell lysates were pre-cleared with Protein A-Sepharose beads for 1 h at room temperature and then immunoprecipitated with anti-Bax sc-493 (Santa Cruz Biotechnology, Dallas, TX, USA) or anti-Acetylated-Lysine #9441 (Cell Signaling, Danvers, MA, USA) antibody overnight at +4 °C. The IP-complexes were collected with Protein A-Sepharose beads (Thermo Fisher Sci., Waltham, MA, USA) for 1 h at room temperature, washed twice with the lysing buffer, and heated at 95 °C for 5 min in a sample buffer (63 mM Tris-HCl, pH 6.8; 1% SDS; 10% glycerol; 5% beta-mercapto-ethanol; 0.01% Br-phenol blue). Proteins were separated by electrophoresis in 10–12% polyacrylamide gel in the presence of 0.1% SDS, transferred onto a membrane (Immobilon P), and probed with appropriate antibodies. As primary antibodies, we used antibodies to Caspase-9 #9504, Caspase-3 #9692, Caspase-3 #9664, Caspase-8 #9429, pAkt #4060, Akt #9272, pErk #4377, pBad # 9296, Bcl-2 #7382, Cytochrome C #11940, Acetylated-Lysine Antibody #9441, GAPDH #2118 (Cell Signaling, Danvers, MA, USA), Bax sc-493 (Santa Cruz Biotechnology, Dallas, TX, USA), and alpha-tubulin T5168 (Sigma-Aldrich, St. Louis, MO, USA). Anti-mouse and anti-rabbit antibodies conjugated with horseradish peroxidase (Sigma-Aldrich, St. Louis, MO, USA) were used as the secondary antibodies. Membrane-bound proteins visualization was achieved using enhanced chemiluminescence (ECL, Amersham Biosciences, Buckinghamshire, UK). Band density analysis was conducted using ImageJ (version 1.53e, NIH, Bethesda, MD, USA). The density values were then normalized to the load control and converted to relative units. The figures represent the mean values from several experiments, with error bars indicating the standard error of the mean (SEM).

### 4.4. Cell Viability Assay (MTT Test)

Cells were seeded at a density of 15,000 per well in a 96-well plate in DMEM with 10% FBS. The cells were then treated with drugs as indicated, and cell viability was assessed using 3-(4,5-dimethylthiazol-2-yl)-2,5-diphenyl tetrazolium bromide (MTT) (Sigma-Aldrich, St. Louis, MO, USA) at a final concentration of 0.5 mg/mL. After a 1-h incubation at 37 °C in a 5% CO2 incubator, the medium was aspirated, and the cells were dissolved in DMSO. The optical density (OD) was measured at 570 nm using a microplate reader Multiscan-EX (Thermo Fisher Scientific, Waltham, MA, USA) with DMSO as a blank solution. The mean of 3–4 independent samples ± standard error of the mean (SEM) were plotted. Statistical significance was determined using Mann–Whitney U-test (* *p* < 0.05, ** *p* < 0.01), with the comparison partners indicated on the graph.

### 4.5. Analysis of In Vitro Drug Interaction

Based on the MTT assay data, a drug interaction coefficient (CDI) was calculated using the formula: CDI = AB/(A × B), where AB represents the ratio of the optical density (OD) at 570 nm for the combination of two drugs to the control group (untreated control), and A or B is the ratio of the single agent group to the control group. Consequently, a CDI value less than 1 indicates drug synergy, equal to 1 indicates an additive effect, and greater than 1 indicates drug antagonism.

### 4.6. Colony Formation Assay

Cells were incubated in the presence or absence of NaBut and/or 2.5 μM etoposide for 24 h. Following drug incubation, the number of cells in each sample was counted and then diluted to seed 200 cells per 30 mm plate. After 10–14 days, the cells were fixed in 100% methanol at room temperature for 30 minutes and then stained for 1 hour with 0.1% crystal violet (Sigma-Aldrich, St. Louis, MO, USA). The stained colonies were counted and compared with an untreated control.

### 4.7. DNA Fragmentation Assay

After appropriate treatments with cytotoxic agents, cells were harvested, washed with PBS, and incubated in a cell lysis buffer (0.25 m EDTA, pH 8.0, 0.25% sarkosyl, and proteinase K (0.1 mg/mL)) for 3 h at 50 °C. Following deproteinization, DNA was incubated in the presence of RNase A (0.5 mg/mL) and 150 mM NaCl for 3 h at 37 °C. DNA fragments were then separated on a 1.5% agarose gel (3 μg/lane). The gels were stained with ethidium bromide and visualized with a UV transilluminator.

### 4.8. Mitochondrial Membrane Potential Assessment (Ψm)

Cells were stained with 50 nM tetramethylrhodamine methyl ester (TMRM, Sigma-Aldrich, St. Louis, MO, USA)) in PBS for 10 min at 37 °C. Changes in mitochondrial membrane potential (Ψm) were determined by assessing the fluorescence intensity of TMRM. This lipophilic, cationic rhodamine derivative is distributed across the cell membrane dependent on membrane potential. The cells were promptly analyzed using a CytoFLEX flow cytometer (Beckman Coulter, Brea, CA, USA), with excitation at 488 nm and detection between 560 and 606 nm.

## 5. Conclusions

Thus, our results demonstrate that the HDAC inhibitor NaBut induces the accumulation of the pro-apoptotic Bax protein in both HDACi-sensitive and HDACi-resistant tumor cells. In HDACi-sensitive cells, NaBut enhances the acetylation of the Ku70 protein, leading to the release of Bax from the inactivating complex in the cytoplasm, the movement of Bax into the mitochondria, an increase in apoptosis, and sensitization to etoposide. In HDACi-resistant cells, the weak acetylation of Ku70 induced by NaBut is insufficient to release the Bax protein from the cytoplasm. Additionally, HDACi-induced activation of the Erk and Akt kinases enhances the phosphorylation of the Bad protein, which further retains the Bax protein in the cytoplasm, preventing its oligomerization and integration into the mitochondrial membrane in HDACi-resistant cells. However, when treated with a combination of HDACi and etoposide, the Bax protein that has accumulated in the cytoplasm due to NaBut treatment is released from the inactivating complex, a result of etoposide-induced MAP kinases inactivation. Consequently, it translocates into the mitochondria, leading to apoptosis.

Despite Bax being inactivated in HDACi-resistant cells, its accumulation in the cytoplasm under NaBut treatment enables the enhancement of the apoptotic response to genotoxic drugs that activate the intrinsic pathway of apoptosis. These results suggest that the combination therapy of HDACis and stress factors that activate the intrinsic apoptotic pathway, such as genotoxic agents, holds promise as a therapeutic strategy for enhancing cytotoxic action in both HDACi-responsive and HDACi-resistant transformed cells.

## Figures and Tables

**Figure 1 ijms-24-15913-f001:**
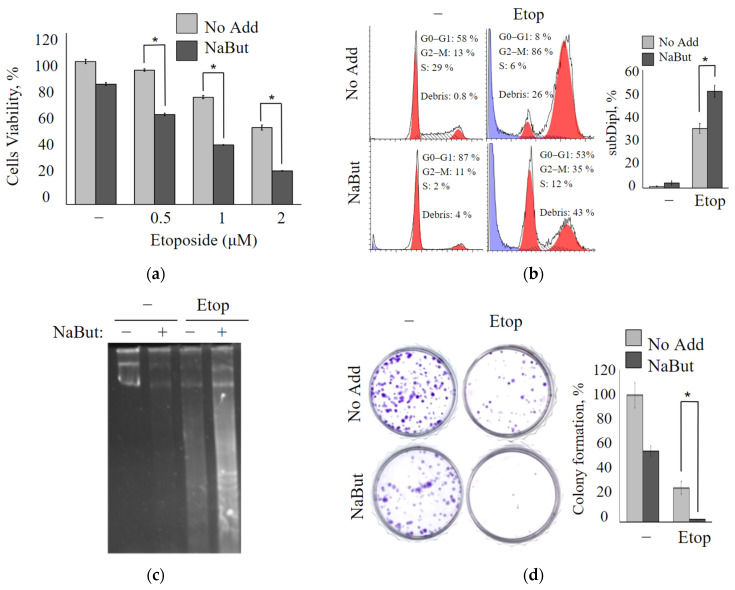
NaBut enhances the cytotoxic effect of etoposide in transformed cells. (**a**) The plot represents the MTT test data for the viability of mERas cells treated with etoposide alone (No Add) or in combination with NaBut (NaBut) for 24 h, as indicated. The y-axis—an absorbance at 570 nm, relative to the control sample (untreated cells). The data represent the mean ± standard error of the mean (SEM) from 3 independent experiments. * *p* < 0.05 by the Mann–Whitney U-test. (**b**) Cell cycle phases were evaluated by flow cytometry analysis in mERas cells following 24 h treatment with 2.5 µM etoposide alone or in combination with 4 mM NaBut. The blue peak on the histogram corresponds to the cells with sub-diploid DNA content. The graphs report the representative distribution, with bars showing the data from 3 independent experiments as a percentage of cells with sub-diploid DNA content. (**c**) Gel electrophoresis of the genomic DNA of mERas cell was performed after treatment with 2.5 μM etoposide alone (−) or in combination with 4 mM NaBut (+) for 24 h. (**d**) A colony formation assay was conducted in mERas cells following treatment with etoposide and/or NaBut, as described in the Section 4. The photo shows the plate with crystal violet-stained colonies, and the plot represents the colony formation efficiency presented as a percentage of control, mean ± SEM (n = 3). * *p* < 0.05 by the Mann–Whitney U-test.

**Figure 2 ijms-24-15913-f002:**
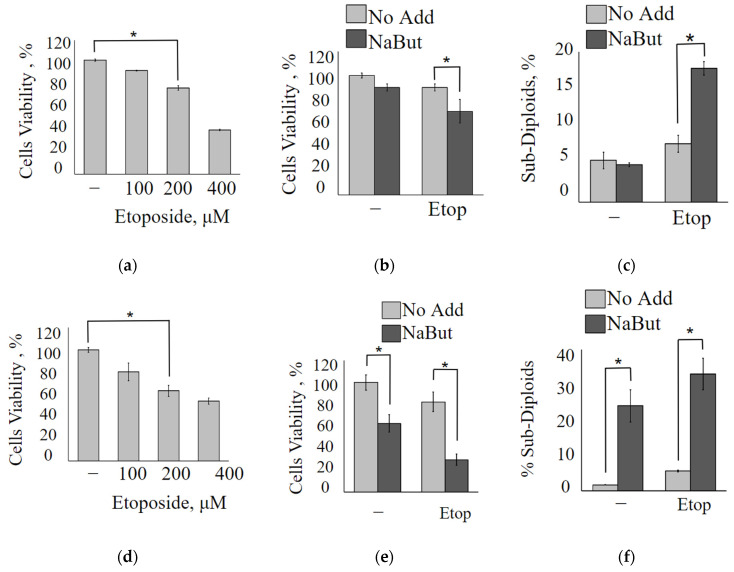
NaBut sensitizes A549 and HCT116 cells to low concentrations of etoposide. The plots represent the MTT data for the viability of A549 (**a**,**b**) or HCT116 cells (**d**,**e**). (**a**,**d**) Cells were treated with increased concentrations of etoposide, as indicated, for 24 h; (**b**,**e**) cells were treated with 5 µM etoposide alone (light bars) or 5 µM etoposide in combination with 4 mM NaBut (dark bars). Results are presented as mean ± SEM of percent of control (untreated cells). (**c**,**f**) The histograms of the flow cytometry data for A549 (**c**) or HCT116 cells (**f**) following the treatment with 5 µM etoposide alone (light bars) or 5 µM etoposide in combination with NaBut (dark bars) 24 h. Bars show the percentage of cells with DNA content below the threshold for cells in G1 (% Sub-Diploids). The MTT and flow cytometry data represent the mean ± SEM from 3 independent experiments. * *p* < 0.05, by Mann–Whitney U-test.

**Figure 3 ijms-24-15913-f003:**
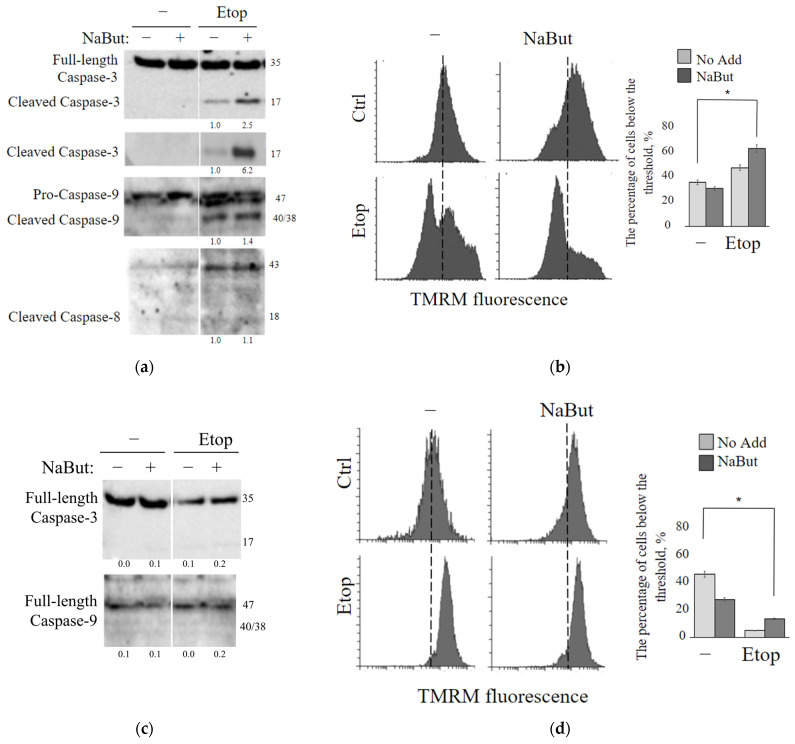
Etoposide induces mitochondria-mediated apoptosis, a process that is mitigated by the overexpression of Bcl-2. (**a**,**c**) Western blot analysis was conducted to detect cleavage of caspases-3, -8, and -9 in mERas (**a**) or mERas-Bcl2 cells (**c**) treated with etoposide (Etop) alone (−) or in combination with NaBut (+) for 24 h. The numbers under the bands represent band densities, calculated using ImageJ© software (version 1.53e). These densities were then scaled to the load control density and normalized against the untreated control (fold changes). (**b**,**d**) Assessment of TMRM staining in mERas (**b**) or mERas-Bcl2 cells (**d**) was conducted on cells that were either left untreated (Ctrl) or treated for 24 h with etoposide alone (−) or in combination with NaBut. TMRM staining was monitored by flow cytometry, and representative cytofluorimetric plots are shown. The graphs represent the percentage of cells below the threshold, marked as a dotted line. The dotted line indicates the position of the peak of fluorescence intensity distribution in the control sample. The asterisk (*) denotes statistical significance (*p* < 0.05), as determined by the U-test.

**Figure 4 ijms-24-15913-f004:**
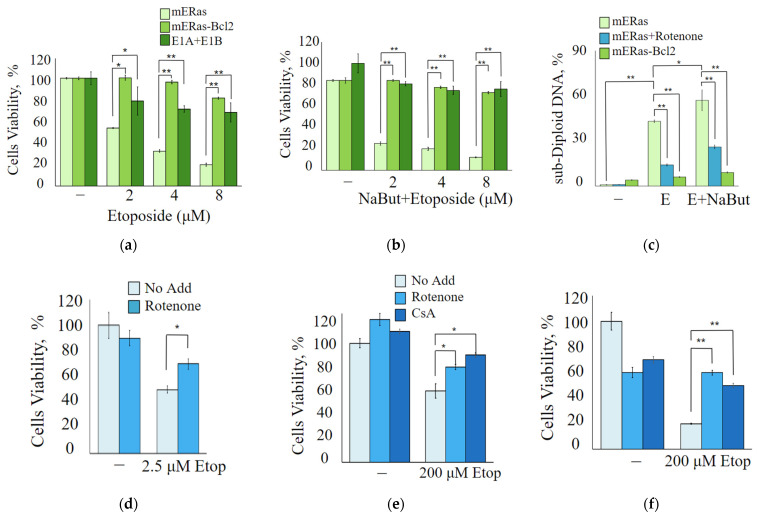
Mitochondrial inhibition attenuates the antiproliferative effect of etoposide. (**a**,**b**) Cell viability was assessed using MTT assay, with results presented as a percentage of control. The data are expressed as mean ± SEM, n = 3. (**a**) mERas cells (light green), mERas-Bcl-2 (green) or E1A + E1B cells (dark green) were either untreated or treated with increased concentrations of etoposide (2–8 μM) alone. (**b**) The same cell lines were treated with etoposide in combination with NaBut for 24 h. (**c**) Histogram of cell flow cytometry data. mERas cells were subjected to different treatments for 24 h: 2.5 μM etoposide alone (E) or etoposide in combination with NaBut (E + NaBut)—light green bars. Simultaneously, the same treatments were conducted with the addition of 50 nM rotenone (mERas + rotenone)—blue bars. Bcl-2 overexpressing mERas cells were also treated with etoposide and NaBut, as described above (mERas-Bcl-2)—green bars. The bars in the histogram represent the proportion of cells in the sub-diploid region of the flow cytometry plots. (**d**–**f**) MTT tests for viability of mERas (**d**), A549 (**e**), and HCT116 cells (**f**) treated with etoposide alone, as indicated (light blue), or in combination with 50 nM rotenone (blue) or 2 μM cyclosporine A (navy) for 24 h. Results are presented as the percentage of viable cells compared with that of untreated control. The MTT and flow cytometry data represent the mean ± SEM from 3 independent experiments. * *p* < 0.05, ** *p* < 0.01 by U-test.

**Figure 5 ijms-24-15913-f005:**
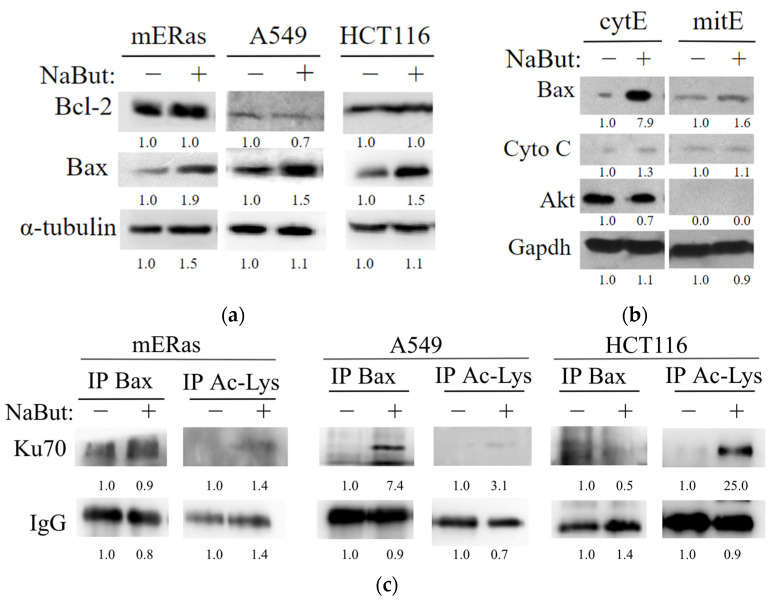
NaBut induces Bax accumulation in cytoplasm of HDACi-resistant cells due to the Bax retention in complexes with Ku70. (**a**) Western blotting of total cell extracts from HDACi-resistant mERas and A549, as well as HDACi-sensitive HCT116 cells, treated with 4 mM NaBut for 24 h. (**b**) Western blotting of the cytoplasmic (cytE) and mitochondrial fractionated extracts (mitE) from HDACi-resistant mERas cells. Western blotting with Akt antibodies served as a marker for cytoplasmic extracts. (**c**) Western blotting with antibodies to Ku70 of proteins co-precipitated with antibodies to Bax or acetylated lysine (Ac-Lys) from HDACi-resistant (mERas and A549) and HDACi-sensitive (HCT116) cells. These cells were either untreated (−) or treated with NaBut for 24 h (+). The numbers under the bands represent band densities calculated by ImageJ© software (version 1.53e), scaled to the load control density, and then normalized against the untreated control (fold changes).

**Figure 6 ijms-24-15913-f006:**
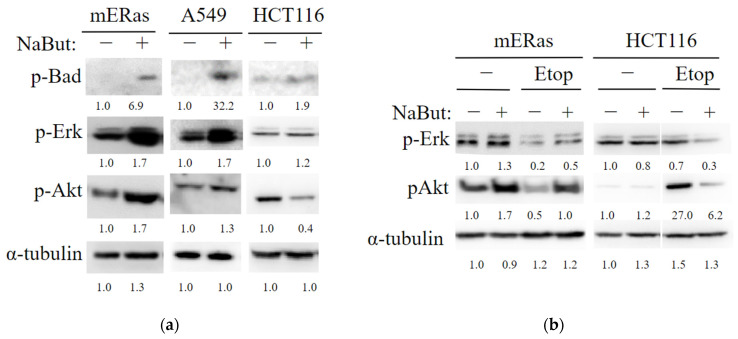
NaBut modulates the activity of MAP kinases differently in HDACi-sensitive and HDACi-resistant cells. (**a**) Western blot analysis of the cell extracts from HDACi-resistant mouse mERas and human tumor A549 cells, as well as HDACi-sensitive HCT116 cells, either untreated (−) or treated with 4 mM NaBut for 24 h (+). (**b**) Western blot analysis of cell extracts from HDACi-resistant mERas and HDACi-sensitive HCT116 cells, either untreated (−) or treated with 5 μM etoposide alone (−) or in combination with 4 mM NaBut (+) for 24 h. The numbers under the bands represent band densities calculated using ImageJ© software (v. 1.53e), scaled to load control density, and then normalized against untreated control (folds).

## Data Availability

The data are not available.

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
