# Peer review of "Sodium Butyrate Enhances the Cytotoxic Effect of Etoposide in HDACi-Sensitive and HDACi-Resistant Transformed Cells"

_ijms, 2023, doi:10.3390/ijms242115913_

Round 1

Reviewer 1 Report

The authors then evaluate the effects of enhanced toxicity of NaBut with EPS on in fibroblast and cancer cells Authors showed NaBut with EPS has enhanced effects on viability and mitochondria-mediated apoptosis on non-tumorigenic or tumor cells. Overall, the author needs to clarify the class of cancer cell death and apoptosis induced by this study to gain insight into the anticancer activity of HDACi. There are several issues that need to be addressed to better support fundamental information on cell induced by etoposide with NaBut.

 Q1. Etoposide with NaBut was investigated by MTT assay in mERas, A549, and HCT116 cells. Please explain why the etoposide and Nabut of concentration of 5uM and 4mM was selected in a549 cells.

 Q2. (In Fig 4a,b) Please explain why the effect on cell survival is different in Bcl-2 cell or Bcl-2 homolog E1B overexpression cells.

Please explain why the concentration of 200 uM etoposide with rotenone and CsA was used.

In Fig 4e, change of cell viability with rotennon/CsA are did not match. Overall, the numbers in the graph need to be changed to (.) instead of (,). There was no information about the rotenone and cyclosporine A used in method.

  Q3. In Fig 6, NaBut modulates the phosphylation of MAP kinases at 24h. In a previous paper, the activation of AKT and ERK was reduced after 48 h. Is it possible to check the expression of p-Bad, p-ERK and p-AKT after 48 h with Etop.

English language fine. No issues detected

Author Response

Reviewer 1

First of all, we would like to thank you for your extensive revision. We have tried to take into account as much of your comments as possible in the time available. Here, we respond to all of them one by one:

The authors then evaluate the effects of enhanced toxicity of NaBut with EPS on in fibroblast and cancer cells Authors showed NaBut with EPS has enhanced effects on viability and mitochondria-mediated apoptosis on non-tumorigenic or tumor cells. Overall, the author needs to clarify the class of cancer cell death and apoptosis induced by this study to gain insight into the anticancer activity of HDACi. There are several issues that need to be addressed to better support fundamental information on cell induced by etoposide with NaBut.

Comment:- Q1. Etoposide with NaBut was investigated by MTT assay in mERas, A549, and HCT116 cells. Please explain why the etoposide and Nabut of concentration of 5uM and 4mM was selected in a549 cells.

Answer:

Human lung carcinoma cells A549 are relatively insensitive to etoposide with respect to mouse fibroblasts, so the etoposide concentration that reduces the viability of A549 cells is more than 200 uM. We analyzed the effect  of a wide range of etoposide concentrations (2-100 µM) in combination with NaBut on the viability of A549 cells and showed that 5 µM is the lowest concentration of etoposide, which in combination with NaBut, has a cytotoxic effect that exceeds the effect of individual drugs.

The concentration of sodium butyrate was selected based on the results of the cell viability test (MTT test). MTT-test data reveal sodium butyrate induces the cell viability decrease in a dose-dependent manner. 2 mM sodium butyrate was unable to affect the cell viability sufficiently, whereas sodium butyrate concentrations of 4 mM and 10 mM reduced the viability of transformed cells with a similar intensity. Therefore, a working concentration of 4 mM sodium butyrate was chosen to induce cell cycle arrest in mouse mERas and human A549 cells, as well as to induce apoptotic death in HCT116 human colon cancer cells.

Comment: Q2. (In Fig 4a,b) Please explain why the effect on cell survival is different in Bcl-2 cell or Bcl-2 homolog E1B overexpression cells.

Answer:

Overexpression of the antiapoptotic Bcl-2 or its functional equivalent adenoviral protein E1B-19K reduces the antiproliferative effects of both etoposide and etoposide with NaBut, but to different extents. E1B-19K is a homolog of the antiapoptotic protein Bcl-2. Despite the sequence similarity between the two proteins, E1B-19K and Bcl-2 use different mechanisms to inactivate the proapoptotic proteins Bax and Bak. Bcl-2 inhibits Bax/Bak activation by binding and inactivating Bid protein which is important to Bax/Bak heterooligomerization at mitochondrial membranes and the induction of apoptosis. E1B-19K, on the other hand, inhibits apoptosis by binding directly to tBid-activated Bax, preventing Bax heterooligomerization and the subsequent downstream events that lead to apoptosis. Probably, the different effect of overexpression of Bcl-2 or its homologue E1B on cell survival at etoposide treatment is a consequence of different mechanisms of the proapoptotic protein Bax inactivation.

We have inserted the corresponding explanation into the text of the article.

Comment: Please explain why the concentration of 200 uM etoposide with rotenone and CsA was used.

Answer:

We are grateful to the reviewer for pointing out that the selected concentration of etoposide in the experiment with MMP inhibitors was not justified.

For the experiment with the inhibitors of the mitochondrial membrane permeability transition we selected the concentration of etoposide 200 uM for more prominent effect. Since we have hypothesized that rotenone and cyclosporine A should cancel the mitochondrion-mediated apoptotic pathway, the concentration of etoposide that significantly reduces cell viability was chosen. 

We have changed the text as follows: ”Inhibitors of mitochondrial membrane permeabilization (cyclosporin A and rotenone) were also used to study the contribution of mitochondria to cell death induced by etoposide. To analyze the modulation of etoposide cytotoxicity by cyclosporin A or rotenone, we used etoposide at a concentration of 200 μM for HCT116 and A549 cells and 2.5 μM for mERas cells, as the lowest concentrations of etoposide that reduce cell viability, respectively. MTT assay data show that chemical inhibitors of mitochondrial membrane permeability transition by cyclosporin A or rotenone reduce the etoposide-induced apoptosis both in HDACi-sensitive HCT116 cells (Fig. 4f) and in HDACi-resistant mERas and A549 cells (Fig. 4d, e).”

Comment: In Fig 4e, change of cell viability with rotennon/CsA are did not match.

Answer:

The differences in the effect of rotenone and cyclosporine A on the abolition of etoposide cytotoxicity can probably be explained by the fact that rotenone, in addition to inhibiting MMPs, is an inhibitor of complex II of the mitochondrial electron transport chain

Comment: Overall, the numbers in the graph need to be changed to (.) instead of (,). There was no information about the rotenone and cyclosporine A used in method.

Answer:

Thank you for your attention, the graphs were updated. The information about rotenone and cyclosporine A was added in the Materials and Methods section. 

To add: “Cells were treated with 4 mM sodium butyrate (NaBut) for 24 hours (Sigma-Aldrich), 2.5-300 μM etoposide (Calbiochem),  50 nM rotenone (Sigma) or  2 μM cyclosporine A (Sigma).”

Comment: Q3. In Fig 6, NaBut modulates the phosphylation of MAP kinases at 24h. In a previous paper, the activation of AKT and ERK was reduced after 48 h. Is it possible to check the expression of p-Bad, p-ERK and p-AKT after 48 h with Etop.

Answer: 

We previously showed that HDACi cause transient activation of Akt and Erk kinases in HDACi-resistant cells, followed by a decrease in their activity after 48 hours. However, the necessity/importance of the MAP kinases activation during the first 24 hours has been demonstrated using a MAPK inhibitor. When MAP kinase activation is prevented by a MAPK inhibitor, HDACi induce apoptosis in HDACi-resistant cells. Therefore, this article focused on events occurring during the first 24 hours of drug action. Thus, MAP-dependent regulation of Bad was analyzed during the first 24 hours as an additional mechanism of apoptotic regulation.

Reviewer 2 Report

Olga O. Gnedina reviewed the molecular mechanisms underlying the enhancement of the cytotoxic effect of etoposide under combined treatment with HDAC inhibitor sodium butyrate (NaBut). They found that NaBut enhances the cytotoxic effect of etoposide in both HDACi-sensitive and HDACi-resistant cells due to the accumulation of the Bax protein and the dissociation of the Ku70-Bax inhibitory complexes. I believe the results are of interest. However, there are several suggestions need to be addressed before publication.

Major revisions:

The authors found that in HDACi-sensitive cells, NaBut enhances the acetylation of the Ku70 protein, which leads to the release of Bax from the inactivating complex in the cytoplasm, the movement of Bax into mitochondria, increased apoptosis, and sensitization to etoposide. In HDACi-resistant cells, weak Ku70 acetylation induced by NaBut is insufficient to release the Bax protein from the cytoplasm. I believe that the results of Bax and Ku70 colocalization will be more convincing via immunofluorescence staining by laser scanning confocal microscopy.

2. Differences of all the results should be analyzed.

Minor comments:

1. HDAC and mERas were first appeared; please provide its full name.

2. Anti-acetyl-lysine antibody was not introduced in the part of 4.3. Immunoblotting.

3. The method of Co-IP was not introduced in the 4. Materials and Methods.

4. The color of the histogram should be consistent, such as Figure 2 and 4.

5. The method of statistical analysis or data analysis was not introduced in the 4. Materials and Methods.

Author Response

Reviewer 2

Thank you for your extensive revision. We have tried to take into account as much of your comments as possible in the time available. Here, we respond to all of them one by one:

Major revisions:

Comment: The authors found that in HDACi-sensitive cells, NaBut enhances the acetylation of the Ku70 protein, which leads to the release of Bax from the inactivating complex in the cytoplasm, the movement of Bax into mitochondria, increased apoptosis, and sensitization to etoposide. In HDACi-resistant cells, weak Ku70 acetylation induced by NaBut is insufficient to release the Bax protein from the cytoplasm. I believe that the results of Bax and Ku70 colocalization will be more convincing via immunofluorescence staining by laser scanning confocal microscopy.

Answer:  We agree with the reviewer that data obtained using laser scanning confocal microscopy after immunofluorescence staining could provide insight into the colocalization of Bax and Ku70 proteins. However, co-immunofluorescence analysis can demonstrate the proximity of proteins, but it cannot prove their physical interactions. Methods such as coimmunoprecipitation or pull-down assays analysis are the common methods for characterizing protein-protein interactions. We limited the presented results only to co-immunoprecipitation data, since the anti-Ku70 antibodies (Cell Signaling) we used are not suitable for IF analysis.

Comment:  Differences of all the results should be analyzed.

Answer:

We are grateful to the reviewer for this helpful comment. All figures were supplemented with the statistical significance of the differences mentioned in the text. For statistical analysis, the Mann-Whitney test (U-test) was used. Comparisons were performed between drug-treated and untreated cells or as indicated in the graph.

In 4.4 Section this clarification was added:

“Statistical significance was determined using a Mann–Whitney U test (* = p < 0.05), the comparison partners are indicated at the graph.”

Minor comments:

  1. HDAC and mERas were first appeared; please provide its full name.

Answer:

We thank the reviewer for pointing out this discrepancy. We have inserted explanations of abbreviations at their first mention in the text.

  1. Anti-acetyl-lysine antibody was not introduced in the part of 4.3. Immunoblotting.

Answer: 

We thank the reviewer for pointing out this omission. We have inserted the catalog number of anti-Acetylated-Lysine Antibody in section 4.3 (Acetylated-Lysine Antibody #9441 (Cell Signaling, Danvers, MA, USA))

  1. The method of Co-IP was not introduced in the 4. Materials and Methods.

Answer: 

We thank the reviewer for noting the lack of the used method description. We have added the description of the IP to the 4.3 section:

“For immunoprecipitation, cells were lysed in a buffer containing 10 mM Tris–HCl, pH 7.4, 150 mM NaCl, 0.5% Nonidet P-40, 1% Triton X-100, 20mM -glycerophosphate, 1mM sodium orthovanadate, 5 mM EGTA, 10 mM sodium fluoride, 1mM phenylmethylsulfonyl fluoride and protease inhibitors cocktail. Before immunoprecipitation,  cell lysates were pre-cleared with Protein A-Sepharose beads for 1 h at room temperature and then immunoprecipitated with anti-Bax sc-493 (Santa Cruz)) or anti-Acetylated-Lysine #9441 (Cell Signaling) antibody overnight at +4 C.  IP-complexes were collected with Protein A-Sepharose beads for 1 h at room temperature and washed twice with the lysing buffer and heated at 95 C for 5 min in a sample buffer (63 mM Tris-HCl pH 6,8; 1% SDS; 10% glycerol; 5% beta-mercapto-ethanol; 0,01% Br-phenol blue).”

  1. The color of the histogram should be consistent, such as Figure 2 and 4.

Answer: 

For better comprehension, we changed the histogram colors of Figure 4, making universal color code for all figure fragments.

  1. The method of statistical analysis or data analysis was not introduced in the 4. Materials and Methods.

Answer: 

We have added a description of the statistical analysis method used in the Materials and Methods section. For statistical analysis, we used the Mann-Whitney test (U test).

In 4.4 Section this clarification was added: “Statistical significance was determined using a Mann–Whitney U test (* = p < 0.05, ** = p < 0.01), the comparison partners are indicated at the graph.”

Reviewer 3 Report

The paper sounds interesting, but first and foremost, the presentation of the data must be improved. Figure 1a: The chart should be presented so that the control is set at 100%. This will be much more readable for the reader. The other parts of the charts b, c, and d should also be presented in bar charts and should show average values, standard deviations, and statistical significances. Figure 2: Similarly, the MTT control should be marked as 100%. The legend of this figure is completely incomprehensible and needs to be corrected. Figure 3: Densitometric analyses and flow cytometry charts, along with standard deviations, should be shown in Figure 4. The legend of this figure definitely needs correction as it is very difficult to understand; controls should be shown as 100%. Figure 5: Is it possible to present densitometric analysis and statistical analyses?

English is generally OK, however, some descriptions, especially the legends of the figures, are too convoluted.

Author Response

Thank you for your comments and suggestions. We have revised the manuscript according to the reviewers comments.

Comment: The paper sounds interesting, but first and foremost, the presentation of the data must be improved. - - Figure 1a: The chart should be presented so that the control is set at 100%. This will be much more readable for the reader. 

Answer:

As recommended by the reviewer, we have presented the histograms in Figures 1a with control set at 100%.

Comment: The other parts of the charts b, c, and d should also be presented in bar charts and should show average values, standard deviations, and statistical significances. 

Answer:

Figure 1c shows a typical DNA content distribution of mERas cells treated with etoposide and etoposide/NaBut to demonstrate the accumulation of a peak with a sub-diploid DNA. A histogram of the average values of 4 independent experiments is shown in Figure 4c. At the reviewer's recommendation, standard errors of the mean and statistical significance were added to the histogram.

Comment:

- Figure 2: Similarly, the MTT control should be marked as 100%. The legend of this figure is completely incomprehensible and needs to be corrected. 

Answer:

As recommended by the reviewer, we have presented the histograms in Figures 2 with control set at 100%.

The caption for Figure 2 has been changed as:

“NaBut sensitises A549 and HCT116 cells to low concentrations of etoposide. The plots represent the MTT data for the viability of A549  (a, b) or HCT116 cells (d, e).  (a, d)  Cells were treated with increased concentrations of etoposide as indicated for 24 h;  (b, e) cells were treated with 5 µM etoposide alone (light bars) or 5 µM etoposide combined with 4 mM NaBut (dark bars). Results are presented as mean ± SEM of percent of control (untreated cells).  (c, f) The histograms of the flow cytometry data for A549 (c) or HCT116 cells (f) following 24 h treatment with 5 µM etoposide  alone (light bars) or 5 µM etoposide combined with NaBut (dark bars). Bars show the percentage of cells with DNA content below the threshold for cells in G1 (% Sub-Diploids). The MTT and flow cytometry data represent the mean ± SEM from 3 independent experiments. * = P < 0.05,  by Mann–Whitney U-test.”

Comment:

- Figure 3: Densitometric analyses and flow cytometry charts, along with standard deviations, should be shown in Figure 4. The legend of this figure definitely needs correction as it is very difficult to understand; controls should be shown as 100%. 

Answer:

The caption for Figure 4 has been rewritten as:

“Mitochondrial inhibition attenuates the antiproliferative effect of etoposide. (a, b)  Cell viability was measured using MTT assay. Results are presented as percent of control, mean ± SEM, n = 3. mERas cells (light green), mERas-Bcl-2 (green) or  E1A+E1B cells (dark green) were left untreated or treated with increased concentration of etoposide (2-8 μM) alone (a) or in combination with NaBut (b) for 24 h.  (с) Histogram of cell flow cytometry data. mERas cells were treated with 2.5 μM etoposide alone (E) or in combination with NaBut (E+NaBut) for 24 h (mERas, light green). mERas cells were treated as described above, but in the presence of 50 nM rotenone (mERas+rotenone, blue). Bcl-2 overexpressing mERas cells were treated with etoposide and NaBut as described above (mERas-Bcl-2, green). The bars show the proportion of cells in the sub-diploid region of the flow cytometry plots. (d-f) MTT tests for viability of mERas (d), A549 (e) and HCT116 cells (f) treated with etoposide alone as indicated (light blue) or in combination with 50 nM rotenone (blue) or  2 μM cyclosporine A (navy) for 24 h. Results are presented as the percentage of viable cells compared with untreated control. The MTT and flow cytometry data represent the mean ± SEM from 3 independent experiments. * = P < 0.05, ** = P < 0.01 by U-test.

As recommended by the reviewer, we have presented the histograms in Figures 4 with control set at 100%.

We are grateful to the reviewer for this helpful comment. All figures were supplemented with the statistical significance of the differences mentioned in the text. For statistical analysis, the Mann-Whitney test (U-test) was used. Comparisons were performed between drug-treated and untreated cells or as indicated in the graph.

Comment:

Figure 5: Is it possible to present densitometric analysis and statistical analyses?

Answer:

Following the reviewer's advice, we performed densitometric analysis of the immunoblotting data and added these results to Figures 5 and 6.

Reviewer 4 Report

In the paper entitled "Sodium butyrate enhances cytotoxic effect of etoposide in HDACi-sensitive and HDACi-resistant transformed cells", the authors present the enhacing potential of sodium butyrate towards etoposide's antiproliferative activity in both HDACi-sensitive and resistant cancer cells.

Considering the continuously rising incidence of cancer and the numerous side effects of anticancer drugs, I believe that the idea of the research conducted by Gnedina et al. is significant and should be shared  with other researchers in the field. 

However, the paper still needs some minor changes before acceptance in IJMS:

- Title: I recommend changing it to "Sodium butyrate enhances the cytotoxic effect of etoposide in HDACi-sensitive and HDACi-resistant transformed cells";

- Introduction: a paragraph about the role of HDAC in cancer cell proliferation should be introduced, in order to make the transfer to their inhibitors (HDACi) more smooth; also, another paragraph about the HDAC inhibitory effect (and mechanism) of NaBut should be inserted, since after reading the introduction, I would expect it to be an authorised HDACi, wich is not;

- References: bibliography must be more up to date, especially in the cancer field (more than 50% of the references cited are 10 to 20 years old); also, check ref. no. 33 (the year of publication is in Bold face) and the journal's indications regarding citation.

Minor English language changes required

Author Response

Reviewer 4

Thank you for your comments and suggestions. We have revised the manuscript according to the reviewers comments:

Considering the continuously rising incidence of cancer and the numerous side effects of anticancer drugs, I believe that the idea of the research conducted by Gnedina et al. is significant and should be shared  with other researchers in the field. 

However, the paper still needs some minor changes before acceptance in IJMS:

Comment: Title: I recommend changing it to "Sodium butyrate enhances the cytotoxic effect of etoposide in HDACi-sensitive and HDACi-resistant transformed cells";

Answer:

We are grateful to the reviewer for correcting the title. 

Comment: Introduction: a paragraph about the role of HDAC in cancer cell proliferation should be introduced, in order to make the transfer to their inhibitors (HDACi) more smooth; also, another paragraph about the HDAC inhibitory effect (and mechanism) of NaBut should be inserted, since after reading the introduction, I would expect it to be an authorised HDACi, wich is not;

Answer:

The introduction was reformatted to be more informative and consistent, two new paragraphs were added:

“Histone deacetylases (HDAC) catalyze removal of the acetyl groups from modified lysine residues located in the N-terminal tail region of the core histones H2A, H2B, H3 and H4. In addition, histone deacetylases could use as substrate not only histone proteins, but other cell proteins, including p53, NFkB, Ku70 etc. HDAC inhibitors induce the hyperacetylation of histones and the chromatin relaxation. The facilitated access of transcriptional factors to the DNA leads to the reactivation of suppressed gene expression.

In this work, we used sodium butyrate (NaBut), a naturally occurring short-chain fatty acid that is a by-product of carbohydrate metabolism in the gut. NaBut is one of the most widely studied HDACi’s, it inhibits histone deacetylases of class I and IIA and its effect is often mediated through Sp1/Sp3 transcription factor.”

Comment: References: bibliography must be more up to date, especially in the cancer field (more than 50% of the references cited are 10 to 20 years old);  also, check ref. no. 33 (the year of publication is in Bold face) and the journal's indications regarding citation.

Answer:

We are grateful to the reviewer for drawing our attention to the broken citation style. We have corrected the citation style in accordance with the journal’s indications. Also, references were partially changed to be more actual.

Round 2

Reviewer 2 Report

no

Author Response

Thank you.